# A theory on the absence of spurious solutions for nonconvex and nonsmooth optimization

**C. Josz**
EECS, UC Berkeley
cedric.josz@gmail.com

**Y. Ouyang**
IEOR, UC Berkeley
ouyangyii@gmail.com

**R. Y. Zhang**
IEOR, UC Berkeley
ryz@berkeley.edu

**J. Lavaei**
IEOR, UC Berkeley
lavaei@berkeley.edu

**S. Sojoudi**
EECS, UC Berkeley
sojoudi@berkeley.edu

## Abstract

We study the set of continuous functions that admit no spurious local optima (i.e. local minima that are not global minima) which we term *global functions*. They satisfy various powerful properties for analyzing nonconvex and nonsmooth optimization problems. For instance, they satisfy a theorem akin to the fundamental uniform limit theorem in the analysis regarding continuous functions. Global functions are also endowed with useful properties regarding the composition of functions and change of variables. Using these new results, we show that a class of nonconvex and nonsmooth optimization problems arising in tensor decomposition applications are global functions. This is the first result concerning nonconvex methods for nonsmooth objective functions. Our result provides a theoretical guarantee for the widely-used $\ell_1$ norm to avoid outliers in nonconvex optimization.

## 1 Introduction

A recent branch of research in optimization and machine learning consists in proving that simple and practical algorithms can solve nonconvex optimization problems. Applications include, but are not limited to, neural networks [40, 44], dictionary learning [1, 2], deep learning [39, 50], mixed linear regression [49, 43], and phase retrieval [46, 21]. In this paper, we focus our attention on matrix completion/sensing [30, 24, 38] and tensor recovery/decomposition [5, 4, 31, 35]. Matrix completion/sensing aims to recover an unknown positive semidefinite matrix $M$ of known size $n$ and rank $r$ from a finite number of linear measurements modeled by the expression $\langle A_i, M \rangle :=$ trace$(A_i M)$, $i = 1, \ldots, m$, where the symmetric matrices $A_1, \ldots, A_m$ of size $n$ are known. It is assumed that the measurements contain noise which can modeled as $b_i := \langle A_i, M \rangle + \epsilon_i$ where $\epsilon_i$ is a realization of a random variable. When the noise is Gaussian, the maximum likelihood estimate of $M$ can be recast as the nonconvex optimization problem

$$\inf_{M \succcurlyeq 0} \quad \sum_{i=1}^{m} \left( \langle A_i, M \rangle - b_i \right)^2 \quad \text{subject to} \quad \text{rank}(M) = r \tag{1}$$

where $M \succcurlyeq 0$ stands for positive semidefinite. One can remove the rank constraint and obtain a convex relaxation. It can then be solved via semidefinite programming after the reformulation of the objective function in a linear way. However, the computational complexity of the resulting problem is high, which makes it impractical for large-scale problems. A popular alternative is due to Burer and Monteiro [18, 12]:

$$\inf_{X \in \mathbb{R}^{n \times r}} \quad \sum_{i=1}^{m} \left( \langle A_i, X X^T \rangle - b_i \right)^2 \tag{2}$$

This nonlinear *Least-Squares* (LS) problem can be solved efficiently and on a large-scale with the Gauss-Newton method for instance. It has received a lot of attention recently due to the discovery in [30, 10] stating that the problem admits no spurious local minima (i.e. local minima that are not global minima) under certain conditions. These require adding a regularizer and satisfying the restricted isometry property (RIP) [20]. We raise the question of whether this also holds in the case of Laplacian noise, which is a better model to account for outliers in the data. The maximum likelihood estimate of $M$ can be converted to the *Least-Absolute Value* (LAV) optimization problem

$$\inf_{X\in\mathbb{R}^{n\times r}} \sum_{i=1}^{m} \left|\langle A_i, XX^T\rangle - b_i\right|. \tag{3}$$

The nonlinear problem can be solved efficiently using nonconvex methods (for some recent work, see [36]). For example, one may adopt the famous reformulation technique for converting $\ell_1$ norms to linear functions subject to linear inequalities to cast the above problem as a smooth nonconvex quadratically-constrained quadratic program [13]. However, the analysis of this result has not been addressed in the literature - all ensuing papers (e.g. [29, 52, 8]) on matrix completion since the aforementioned discovery deal with smooth objective functions.

Consider $y \in \mathbb{R}^n$ and assume $r = 1$. On the one hand, in the fully observable case[1] with $M = yy^T$, the above nonconvex LS problem (2) consists in solving

$$\inf_{x\in\mathbb{R}^n} \sum_{i,j=1}^{n} (x_i x_j - y_i y_j - \epsilon_{i,j})^2 \tag{4}$$

for which there are no spurious local minima with high probability when $\epsilon_{i,j}$ are i.i.d. Gaussian variables [30]. On the other hand, in the full observable case, the LAV problem (3) aims to solve

$$\inf_{x\in\mathbb{R}^n} \sum_{i,j=1}^{n} |x_i x_j - y_i y_j - \epsilon_{i,j}|. \tag{5}$$

Although the LS problem has nice properties with Gaussian noise, we observe that stochastic gradient descent (SGD) fails to recover the matrix $M = yy^T$ in the presence of large but sparse noise. In contrast, SGD can perfectly recover the matrix by solving the LAV problem even when the sparse noise $\epsilon_{i,j}$ has a large amplitude. Figures 1a and 1b show our experiments for $n = 20$ and $n = 50$ with the number of noisy elements ranging from $0$ to $n^2$. See Appendix 5.1 for our experiment settings.

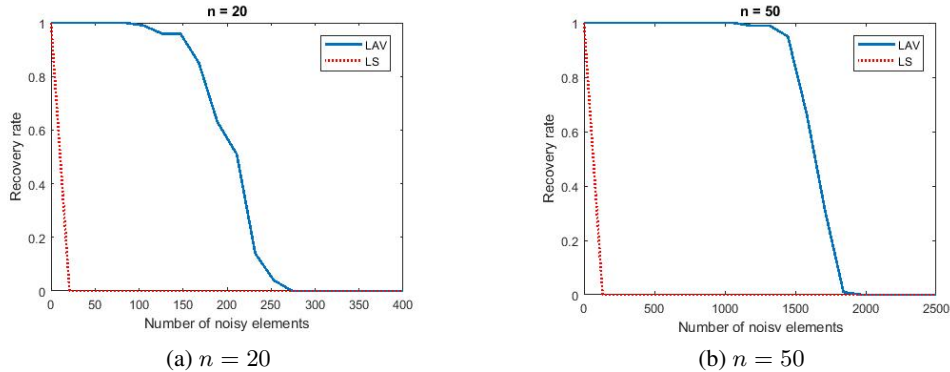

(a) $n = 20$
(b) $n = 50$

Figure 1: Experiments with sparse noise

Upon this LAV formulation hinges the potential of nonconvex methods to cope with sparse noise and with Laplacian noise. There is no result on the analysis of the local solutions of this nonsmooth problem in the literature even for the noiseless case. This could be due to the fact that the optimality conditions for the smooth reformulated version of this problem in the form of quadratically-constrained quadratic program are highly nonlinear and lead to an exponential number of scenarios.

As such, the goal of this paper is to prove the following proposition, which as the reader will see, is a significant hurdle. It addresses the matrix noiseless case and more generally the case of a tensor of order $d \in \mathbb{N}$.

**Proposition 1.1.** *The function $f_1 : \mathbb{R}^n \longrightarrow \mathbb{R}$ defined as*

$$f_1(x) := \sum_{i_1,\ldots,i_d=1}^{n} |x_{i_1} \ldots x_{i_d} - y_{i_1} \ldots y_{i_d}| \tag{6}$$

*has no spurious local minima.*

A direct consequence of Proposition 1.1 is that one can perform the rank-one tensor decomposition by minimizing the function in Proposition 1.1 using a local search algorithm (e.g. [19]). Whenever the algorithm reaches a local minimum, it is a globally optimal solution leading to the desired decomposition. Existing proof techniques, e.g. [29, 30, 24, 38, 5, 4, 31, 35], are not directly useful for the analysis of the nonconvex and nonsmooth optimization problem stated above. In particular, results on the absence of spurious local minima neural networks with a Rectified Linear Unit (ReLU) activation function pertain to smooth objective functions (e.g. [48, 14]). The Clarke derivative [22, 23] provides valuable insight (see Lemma 3.1) but it is not conclusive. In order to pursue the proof (see Lemma 3.2), we propose the new notion of global function. Unlike the previous approaches, it does not require one to exhibit a direction of descent. After some successive transformations, we reduce the problem to a linear program. It is then obvious that there are no spurious local minima. Incidentally, global functions provide a far simpler and shorter proof to a slightly weaker result, that is to say, the absence of spurious strict local minima. It eschews the Clarke derivative all together and instead considers a sequence of converging differentiable functions that have no spurious local minima (see Proposition 3.1). In fact, this technique also applies if we substitute the $\ell_1$ norm with the $\ell_\infty$ norm (see Proposition 3.2).

The paper is organized as follows. Global functions are examined in Section 2 and their application to tensor decomposition is discussed in Section 3. Section 4 concludes our work. The proofs may be found in the supplementary material (Section 5 of the supplementary material).

## 2 Notion of global function

Given an integer $n$, consider the Euclidian space $\mathbb{R}^n$ with norm $\|x\|_2 := \sqrt{\sum_{i=1}^{n} x_i^2}$ along with a subset $S \subset \mathbb{R}^n$. The next two definitions are classical.

**Definition 2.1.** *We say that $x \in S$ is a global minimum of $f : S \longrightarrow \mathbb{R}$ if for all $y \in S \setminus \{x\}$, it holds that $f(x) \leqslant f(y)$.*

**Definition 2.2.** *We say that $x \in S$ is a local minimum (respectively, strict local minimum) of $f : S \longrightarrow \mathbb{R}$ if there exists $\epsilon > 0$ such that for all $y \in S \setminus \{x\}$ satisfying $\|x - y\|_2 \leqslant \epsilon$, it holds that $f(x) \leqslant f(y)$ (respectively, $f(x) < f(y)$).*

We introduce the notion of global functions below.

**Definition 2.3.** *We say that $f : S \longrightarrow \mathbb{R}$ is a global function if it is continuous and its local minima are all global minima. Define $\mathcal{G}(S)$ as the set of all global functions on $S$.*

In the following, we compare global functions with other classes of functions in the literature, particularly those that seek to generalize convex functions.

When the domain $S$ is convex, two important proper subsets of $\mathcal{G}(S)$ are the sets of convex and strict quasiconvex functions. Convex functions (respectively, strict quasiconvex [27, 26]) are such that $f(\lambda x + (1 - \lambda)y) \leqslant \lambda f(x) + (1 - \lambda)f(y)$ (respectively, $f(\lambda x + (1 - \lambda)y) < \max\{f(x), f(y)\}$) for all $x, y \in S$ (with $x \neq y$) and $0 < \lambda < 1$. To see why these are proper subsets, notice that the cosine function on $[0, 4\pi]$ is a global function that is neither convex nor strict quasiconvex. In dimension one, global and strict quasiconvex functions are very closely related. Indeed, when the domain is convex and compact (i.e. an interval $[a, b]$ where $a, b \in \mathbb{R}$), it can be shown that a function is strict quasiconvex if and only if it is global and has a unique global minimum. However, this is not true in higher dimensions, as can be seen in Figure 4b in Appendix 5.2, or in the existing literature, i.e. in

[25] or in [9, Figure 1.1.10]. It is also not true in dimension one if we remove the assumption that the domain is compact (consider $f(x) := (x^2 + x^4)/(1 + x^4)$ defined on $\mathbb{R}$ and illustrated in Figure 4a in Appendix 5.2).

When the domain $S$ is not necessarily convex, a proper subset of $\mathcal{G}(S)$ is the set of star-convex functions. For a star-convex function $f$, there exists $x \in S$ such that $f(\lambda x + (1-\lambda)y) \leqslant \lambda f(x) + (1-\lambda)f(y)$ for all $y \in S \setminus \{x\}$ and $0 < \lambda < 1$. Again, the cosinus function on $[0, 4\pi]$ is a global function that is not star-convex. Another interesting proper subset of $\mathcal{G}(S)$ is the set of functions for which, informally, given any point, there exists a strictly decreasing path from that point to a global minimum. This property is discussed in [47, P.1] (see also [28]) to study the landscape of loss functions of neural networks. Formally, the property is that for all $x \in S$ such that $f(x) > \inf_{y \in S} f(y)$, there exists a continuous function $g : [0, 1] \longrightarrow S$ such that $g(0) = x$, $g(1) \in \operatorname{argmin}\{f(y) \mid y \in S\}$, and $t \in [0, 1] \longmapsto f(g(t))$ is strictly decreasing (i.e. $f(g(t_1)) > f(g(t_2))$ if $0 \leqslant t_1 < t_2 \leqslant 1$). Not all global functions satisfy this property, as illustrated by the function in Figure 4a. For instance, there exists no strictly decreasing path from $x = -3$ to the global minimizer 0. However, in the funtion in Figure 4b in Appendix 5.2, there exists a strictly decreasing path from any point to the unique global minimizer. One could thus think that if $S$ is compact, or if $f$ is coercive, then one should always be able to find a strictly decreasing path. However, there need not exist a strictly decreasing path in general. Consider for example the function defined on $([-1, 1] \setminus \{0\}) \times [-1, 1]$ as follows

$$
f(x_1, x_2) := \begin{cases} -4|x_1|^3(1 - x_2)\left(\sin\left(-\frac{1}{|x_1|}\right) + 1\right) & \text{if} \quad 0 \leqslant x_2 \leqslant 1, \\[2mm] \left\{12|x_1|^3\left(\sin\left(-\frac{1}{|x_1|}\right) + 1\right) - 2\right\} x_2^3 + \\[2mm] \left\{20|x_1|^3\left(\sin\left(-\frac{1}{|x_1|}\right) + 1\right) - 3\right\} x_2^2 + & \text{if} \; -1 \leqslant x_2 < 0. \\[2mm] 4|x_1|^3\left(\sin\left(-\frac{1}{|x_1|}\right) + 1\right) x_2 - 4|x_1|^3\left(\sin\left(-\frac{1}{|x_1|}\right) + 1\right) \end{cases}
$$

The function and its differential can readily be extended continuously to $[-1, 1] \times [-1, 1]$. This is illustrated in Figure 6a in Appendix 5.2. This yields a smooth[2] global function for which there exists no strictly decreasing path from the point $x = (0, 1/2)$ to a global minimizer (i.e. any point in $[-1, 1] \times \{-1\}$). We find this to be rather counter-intuitive. To the best of our knowledge, no such function has been presented in past literature. Hestenes [32] considered the function defined on $[-1, 1] \times [-1, 1]$ by $f(x_1, x_2) := (x_2 - x_1^2)(x_2 - 4x_1^2)$ (see also [9, Figure 1.1.18]). It is a global function for which the point $x = (0, 0)$ (which is not a global minimizer) admits no direction of descent, i.e. $d \in \mathbb{R}^2$ such that $t \in [0, 1] \longmapsto f(x + td)$ is strictly decreasing. However, it does admit a strictly decreasing path to a global minimizer, i.e. $t \in [0, 1] \longmapsto (\frac{\sqrt{10}}{4}t, t^2)$, along which the objective equals $-\frac{9}{16}t^4$. This is unlike the function exhibited in Figure 6a. As a byproduct, our function shows that the generalization of quasiconvexity to non-convex domains described in [6, Chapter 9] is a proper subset of global functions. This generalization was proposed in [41] and further investigated in [7, 33, 34, 15, 16, 17]. It consists in replacing the segment used to define convexity and quasiconvexity by a continuous path.

Finally, we note that there exists a characterization of functions whose local minima are global, without requiring continuity as in global functions. It is based on a certain notion of continuity of sublevel sets, namely lower-semicontinuity of point-to-set mappings [51, Theorem 3.3]. We will see below that continuity is a key ingredient for obtaining our results. We do not require more regularity precisely because one of our goals is to study nonsmooth functions. Speaking of which, observe that global functions can be nowhere differentiable, contrary to convex functions [11, Theorems 2.1.2 and 2.5.1]. Consider for example the global function defined on $]0, 1[ \times ]0, 1[$ by $f(x_1, x_2) := |2x_2 - 1| \sum_{n=0}^{+\infty} s(2^n x_1)/2^n$ where $s(x) := \min_{n \in \mathbb{N}} |x - n|$ is the distance to nearest integer. For any fixed $x_2 \neq 0$, the function $x_1 \in [0, 1] \longmapsto f(x_1, x_2)/|x_2|$ is the Takagi curve [45, 3, 37] which is nowhere differentiable. It can easily be deduced that the bivariate function is nowhere differentiable. This is illustrated in Figure 6b.

In the following, we review some of the properties of global functions. Their proofs can be found in the appendix. We begin by investigating the composition operation.

**Proposition 2.1** (Composition of functions). *Consider $f : S \longrightarrow \mathbb{R}$. Let $\phi : f(S) \longrightarrow \mathbb{R}$ denote a strictly increasing function where $f(S)$ is the range of $f$. It holds that $f \in \mathcal{G}(S)$ if and only if $\phi \circ f \in \mathcal{G}(S)$.*

However, the set of global functions is not closed under composition of functions in general. For instance, $f(x) := |x|$ and $g(x) := \max(-1, |x| - 2)$ are global functions on $\mathbb{R}$, but $f \circ g$ is not global function on $\mathbb{R}$.

**Proposition 2.2** (Change of variables). *Consider $f : S \longrightarrow \mathbb{R}$, a subset $S' \subset \mathbb{R}^n$, and a homeomorphism $\varphi : S \longrightarrow S'$ (i.e. continuous bijection with continuous inverse). It holds that $f \in \mathcal{G}(S)$ if and only if $f \circ \varphi^{-1} \in \mathcal{G}(S')$.*

Next, we consider what happens if we have a sequence of global functions. Figure 2a shows that the sequence of global functions (red dotted curves) pointwise converges to a function with a spurious local minimum (blue curve). Figure 2b shows that uniform convergence also does not preserve the property of being a global function: all points on the middle part of the limit function (blue curve) are spurious local minima. However, it suggests that uniform convergence preserves a slightly weaker property than being a global function. Intuitively, the limit should behave like a global function except that it may have "flat" parts. We next formalize this intuition. To do so, we consider the notions of global minimum, local minimum, and strict local minimum (Definition 2.1 and Definition 2.2), which apply to points in $\mathbb{R}^n$, and generalize them to subsets of $\mathbb{R}^n$. We will borrow the notion of neighborhood of a set (*uniform neighborhood* to be precise, see Definition 2.5).

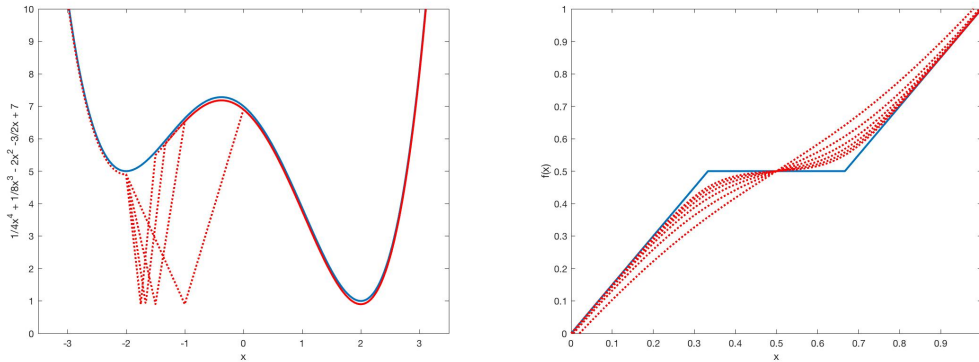

(a) Pointwise convergence          (b) Uniform convergence

Figure 2: Convergence of a sequence of global functions

**Definition 2.4.** *We say that a subset $X \subset S$ is a global minimum of $f : S \longrightarrow \mathbb{R}$ if $\inf_X f \leqslant \inf_{S \setminus X} f$.*

We note in passing the following two propositions. We will use them repeatedly in the next section. The proofs are omitted as they follow directly from the definitions.

**Proposition 2.3.** *Assume that the following statements are true:*

    *1. $X \subset S$ is a global minimum of $f$;*

    *2. $f \in \mathcal{G}(X)$;*

    *3. $f$ does not have any local minima on $S \setminus X$.*

*Then, $f \in \mathcal{G}(S)$.*

Note that the first assumption is needed; otherwise the function may not be global because it could take a smaller value at a non local min outside X (possible when S is unbounded).

**Proposition 2.4.** *If $f : S \longrightarrow \mathbb{R}$ is a global function on global minima $(X_\alpha)_{\alpha \in A}$ for some index set $A$, then it is a global function on $\bigcup_{\alpha \in A} X_\alpha$.*

We proceed to generalize the definition of local minimum.

**Definition 2.5.** *We say that a compact subset $X \subset S$ is local minimum (respectively, strict local minimum) of $f : S \longrightarrow \mathbb{R}$ if there exists $\epsilon > 0$ such that for all $x \in X$ and for all $y \in S \setminus X$ satisfying $\|x - y\|_2 \leqslant \epsilon$, it holds that $f(x) \leqslant f(y)$ (respectively, $f(x) < f(y)$).[3]*

The above definitions are distinct from the notion of valley proposed in [47, Definition 1]. The latter is defined as a connected component[4] of a sublevel set (i.e. $\{x \in S \mid f(x) \leqslant \alpha\}$ for some $\alpha \in \mathbb{R}$). Local minima and strict local minima need not be valleys, and vice-versa. One may easily check that when the set is a point, i.e. $X = \{x\}$ with $x \in S$, the two definitions above are the same as the previous definitions of minimum (Definition 2.1 and Definition 2.2). They are therefore consistent. It turns out that the notion of global function (Definition 2.3) does not change when we interpret it in the sense of sets. We next verify this claim.

**Proposition 2.5** (Consistency of Definition 2.3). *Let $f : S \longrightarrow \mathbb{R}$ denote a continuous function. All local minima are global minima in the sense of points if only if all local minima are global minima in the sense of sets.*

We are ready to define a slightly weaker notion than being a global function.

**Definition 2.6.** *We say that $f : S \longrightarrow \mathbb{R}$ is a weakly global function if it is continuous and if all strict local minima are global minima in the sense of sets.*

The generalization from points to sets in the definition of a minimum is justified here, as can be seen in Figure 3. All strict local minima are global minima in the sense of points. However, $X = [a, b]$ with $a \approx -2.6$ and $b = -1$ is a strict local minimum that is not a global minimum. Indeed, $\inf_X f = 6 > 1 = \inf_{\mathbb{R} \setminus X} f$. Hence, the function is not weakly global.

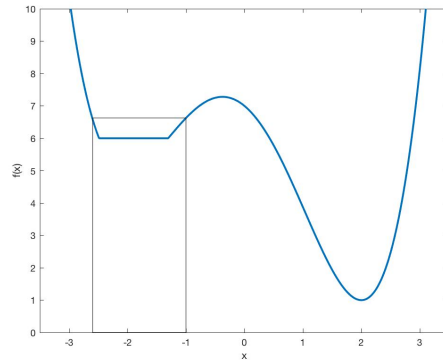

Figure 3: All strict local minima are global minima in the sense of points but not in the sense of sets.

We next make the link with the intuition regarding the flat part in Figure 2b.

**Proposition 2.6.** *If $f : S \longrightarrow \mathbb{R}$ is a weakly global function, then it is constant on all local minima that are not global minima.*

We are interested in functions that are potentially defined on all of $\mathbb{R}^n$ (i.e. unconstrained optimization) or on subsets $S \subset \mathbb{R}^n$ that are not necessarily compact (i.e. general constrained optimization). We therefore need to borrow a slightly more general notion than uniform convergence [42, page 95, Section 3].

**Definition 2.7.** *We say that a sequence of continuous functions $f_k : S \longrightarrow \mathbb{R}, k = 1, 2, \ldots,$ converges compactly towards $f : S \longrightarrow \mathbb{R}$ if for all compact subsets $K \subset S$, the restrictions of $f_k$ to $K$ converge uniformly towards the restriction of $f$ to $K$.*

We are now ready to state a result regarding the convergence of a sequence of global functions and an important property that is preserved in the process.

**Proposition 2.7** (Compact convergence). *Consider a sequence of functions $(f_k)_{k\in\mathbb{N}}$ and a function $f$, all from $S \subset \mathbb{R}^n$ to $\mathbb{R}$. If*

$$f_k \longrightarrow f \quad \text{compactly} \tag{7}$$

*and if $f_k$ are global functions on $S$, then $f$ is a weakly global function on $S$.*

Note that the proofs in this section are not valid if we replace the Euclidian space by an infinite-dimensional metric space. Indeed, we have implicitly used the fact that the unit ball is compact in order for the uniform neighborhood of a minimum to be compact.

## 3 Application to tensor decomposition

Global functions can be used to prove the following two significant results on nonlinear functions involving $\ell_1$ norm and $\ell_\infty$ norm, as explained below.

**Proposition 3.1.** *The function $f_1 : \mathbb{R}^n \longrightarrow \mathbb{R}$ defined as*

$$f_1(x) := \sum_{i_1,\dots,i_d=1}^{n} |x_{i_1}\dots x_{i_d} - y_{i_1}\dots y_{i_d}| \tag{8}$$

*is a weakly global function; in particular, it has no spurious strict local minima.*

*Proof.* The functions

$$f_p(x) := \sum_{i_1,\dots,i_d=1}^{n} |x_{i_1}\dots x_{i_d} - y_{i_1}\dots y_{i_d}|^p \tag{9}$$

for $p \longrightarrow 1$ with $p > 1$ form a set of global functions that converge compactly towards the function $f_1$. This is illustrated in Figure 5 in Appendix 5.2 for $n = d = 2$ and $y = (1, -3/4)$. The desired result then follows from Proposition 2.7. To see why each $f_p$ is a global function, observe that $f_p$ is differentiable with the first-order optimality condition as follows:

$$\sum_{i_1,\dots,i_{d-1}=1}^{n} x_{i_1}\dots x_{i_{d-1}}(x_{i_1}\dots x_{i_{d-1}}x_i - y_{i_1}\dots y_{i_{d-1}}y_i)|x_{i_1}\dots x_{i_{d-1}}x_i - y_{i_1}\dots y_{i_{d-1}}y_i|^{p-2} = 0$$

for all $i \in \{1,\dots,n\}$. Note that each term in the sum converges towards zero if the expression inside the absolute value converges towards zero, so that the equation is well-defined. Consider a local minimum $x \in \mathbb{R}^n$; then, $x$ must satisfy the above first-order optimality condition. If $y_i = 0$, then the above equation readily yields $x_i = 0$. This reduces the problem dimension from $n$ variables to $n-1$ variables, so without loss of generality we may assume that $y_i \neq 0$, $i = 1,\dots,m$. After a division, observe that the following equation is satisfied

$$\sum_{i_1,\dots,i_{d-1}=1}^{n} |y_{i_1}\dots y_{i_{d-1}}|^p \frac{x_{i_1}\dots x_{i_{d-1}}}{y_{i_1}\dots y_{i_{d-1}}}\left(\frac{x_{i_1}\dots x_{i_{d-1}}}{y_{i_1}\dots y_{i_{d-1}}}t - 1\right)\left|\frac{x_{i_1}\dots x_{i_{d-1}}}{y_{i_1}\dots y_{i_{d-1}}}t - 1\right|^{p-2} = 0$$

for all $t \in \{x_1/y_1,\dots,x_n/y_n\}$. Each term with $x_{i_1}\dots x_{i_{d-1}} \neq 0$ in the above sum is a strictly increasing function of $t \in \mathbb{R}$ since it is the derivative of the strictly convex function

$$g(t) = |x_{i_1}\dots x_{i_{d-1}}t - y_{i_1}\dots y_{i_{d-1}}|^p. \tag{10}$$

The point $x = 0$ is not a local minimum ($y$ is a direction of descent of $f_p$ at 0), and thus $x \neq 0$. As a result, the above sum is a strictly increasing function of $t \in \mathbb{R}$. Hence, it has at most one root, that is to say $t = x_1/y_1 = \dots = x_n/y_n$. Plugging in, we find that $t^d = 1$. If $d$ is odd, then $x = y$ and if $d$ is even, then $x = \pm y$. To conclude, any local minimum $x$ is a global minimum of $f_p$. $\square$

**Proposition 3.2.** $f_\infty : \mathbb{R}^n \longrightarrow \mathbb{R}$ *defined as*

$$f_\infty(x) := \max_{1 \leqslant i_1,\dots,i_d \leqslant n} |x_{i_1}\dots x_{i_d} - y_{i_1}\dots y_{i_d}| \tag{11}$$

*is a weakly global function; in particular, it has no spurious strict local minima.*

*Proof.* The functions $h_p(x) := \left( \sum\limits_{i_1,\ldots,i_d=1}^{n} |x_{i_1} \ldots x_{i_d} - y_{i_1} \ldots y_{i_d}|^p \right)^{\frac{1}{p}}$ for $p \longrightarrow +\infty$ form a set of global functions that converge compactly towards the function $f_\infty$. We know that each $h_p$ is a global function by applying Proposition 2.1 to (9) with the fact that $(\cdot)^{\frac{1}{p}}$ is increasing for nonnegative arguments. $\qquad \square$

Note that the functions in Proposition 3.1 and Proposition 3.2 are *a priori* utterly different, yet both proofs are essentially the same. This highlights the usefulness of the new notion of global functions.

***Remark* 3.1.** *The notion of weakly global functions explains that one can perform tensor decomposition by minimizing the nonconvex and nonsmooth functions in Proposition 3.1 and Proposition 3.2 with a local search algorithm. Whenever the algorithm reports a strict local minimum, it is a globally optimal solution.*

In order to strengthen the conclusion in Proposition 3.1 and to establish the absence of spurious local minima, we propose the following two lemmas. Using Proposition 2.3 and these two lemmas, we arrive at the stronger result stated in Proposition 1.1.

***Lemma* 3.1.** *If $x \in \mathbb{R}^n$ is a first-order stationary point of $f_1$ in the sense of the Clarke derivative, then the following statements hold:*

1. *If $y_i = 0$ for some $i \in \{1, \ldots, n\}$, then $x_i = 0$;*

2. *For all $i_1, \ldots, i_d \in \{1, \ldots, n\}$, it holds that $\frac{x_{i_1} \ldots x_{i_d}}{y_{i_1} \ldots y_{i_d}} \leqslant 1$.*

*Proof.* Similar in spirit to the proof of Proposition 3.1, the ratios $t \in \{x_1/y_1, \ldots, x_n/y_n\}$ for a first-order stationary point must all be the roots of an increasing (set-valued) "staircase function". We then obtain the results by analyzing the relation between the roots and the jump points of the staircase function. See Appendix 5.8 for the complete proof. $\qquad \square$

Note that the above lemma only uses the first-order optimality condition (in the sense of Clarke derivative) without any direction of decent.

***Remark* 3.2.** *One cannot show that there are no spurious local minima with only the first-order optimality condition (in the Clarke derivative sense). In fact, any $x \in \mathbb{R}^n$ satisfying $\sum\limits_{i=1}^{n} |y_i| \frac{x_i}{y_i} = 0$ and $\frac{x_{i_1} \ldots x_{i_d}}{y_{i_1} \ldots y_{i_d}} \leqslant 1$ for all $i_1, \ldots, i_d \in \{1, \ldots, n\}$, is a first-order stationary point, but is not a local minimum.*

***Lemma* 3.2.** *If $y_1 \ldots y_n \neq 0$, define the set*

$$S := \left\{ x \in \mathbb{R}^n \mid \frac{x_{i_1} \ldots x_{i_d}}{y_{i_1} \ldots y_{i_d}} \leqslant 1, \quad \forall \, i_1, \ldots, i_d \in \{1, \ldots, n\} \right\}. \tag{12}$$

*Then, $f_1 \in \mathcal{G}(S)$.*

*Proof.* We provide a sketch here, and the complete proof is deferred to Appendix 5.9. The objective function on $S$ is equal to $f_1(x) = \left( \sum\limits_{i=1}^{n} |y_i| \right)^d - \left( \sum\limits_{i=1}^{n} |y_i| \frac{x_i}{y_i} \right)^d$. Define the set $S' := \left\{ x \in \mathbb{R}^n \mid x_{i_1} \ldots x_{i_d} \leqslant 1, \quad \forall \, i_1, \ldots, i_d \in \{1, \ldots, n\} \right\}$. When $d$ is an odd number, the composition and change of variables properties of global functions (Propositions 2.1 and 2.2) imply that $f_1$ is a global function on $S$ if and only if $f_{\text{odd}}(x) = -\sum_{i=1}^{n} |y_i| x_i \in \mathcal{G}(S')$. Similarly, when $d$ is an even number, $f$ is a global function if and only if $f_{\text{even}}(x) = -\left( \sum_{i=1}^{n} |y_i| x_i \right)^2 \in \mathcal{G}(S')$. For the case when $d$ is odd, we apply the Karush-Kuhn-Tucker conditions to restrict attention to the positive orthant and conclude by showing its association with a linear program. For the case when $d$ is even, we divide the set $S'$ into two subsets: $S' \cap \{x | \sum_{i=1}^{n} |y_i| x_i \geq 0\}$ and $S' \cap \{x | \sum_{i=1}^{n} |y_i| x_i \leq 0\}$. Observe that $f_{\text{even}}(x)$ is a global function on each of the subset by associating each subset with a linear program. Then, Proposition 2.3 establishes the result. $\qquad \square$

The two previous lemmas prove Proposition 1.1; the notion of global function is used to the prove the latter.

# 4    Conclusion

Nonconvex optimization appears in many applications, such as matrix completion/sensing, tensor recovery/decomposition, and training of neural networks. For a general nonconvex function, a local search algorithm may become stuck at a local minimum that is arbitrarily worse than a global minimum. We develop a new notion of global functions for which all local minima are global minima. Using certain properties of global functions, we show that the set of these functions include a class of nonconvex and nonsmooth functions that arise in matrix completion/sensing and tensor recovery/decomposition with Laplacian noise. This paper offers a new mathematical technique for the analysis of nonconvex and nonsmooth functions such as those involving $\ell_1$ norm and $\ell_\infty$ norm.

**Acknowledgments**

This work was supported by the ONR Awards N00014-17-1-2933 ONR and N00014-18-1-2526, NSF Award 1808859, DARPA Award D16AP00002, and AFOSR Award FA9550- 17-1-0163. We wish to thank the anonymous reviewers for their valuable feedback, as well as Chris Dock for fruitful discussions.

## Footnotes

[1]This corresponds to the case where the sensing matrices $A_1, \ldots, A_{n^2}$ have all zeros terms apart from one element which is equal to 1.

[2]In fact, one could make it infinitely differentiable by using the exponential function in the construction, but it is more cumbersome.

[3]Note that the neighborhood of a compact set is always uniform.

[4]A subset $C \subset S$ is connected if it is not equal to the union of two disjoint nonempty closed subsets of $S$. A maximal connected subset (ordered by inclusion) of $S$ is called a connected component.
