[Supplementary Material]

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

# 5 Appendix

## 5.1 Experiment settings

We use SGD to solve the problems (4) and (5) for randomly generated rank-one matrices. In the experiments, each $y$ is generated according to the $n$-dimensional i.i.d. standard Gaussian distribution. The positions of the sparse noise are uniformly selected from all the $n^2$ elements, and each noisy element is replaced by a Gaussian random variable with standard deviation 10. With regard to SGD, we set the learning rate to $0.001$ and momentum to $0.9$. The initial point is a Gaussian random vector.

In our experiments, a successful recovery means that the solution $x$ has a relative error less than $0.1$ compared with the optimal solution $y$. We consider $n = 20$ and $n = 50$ and vary the number of noisy elements from 0 to $n^2$. For each case, we run 100 experiments and report the successful recovery rate. As shown in Figures (1a) and (1b), the LS problem (4) fails to recover the matrix except for the noiseless case. On the other hand, the LAV problem (4) provides perfect recovery in the presence of sparse noise.

## 5.2 Illustrations

This section is composed of Figure 4, Figure 5, and Figure 6.

(a) Global function on $\mathbb{R}$

(b) Global function on $\mathbb{R}^2$

Figure 4: Examples of global functions

Concerning Figure 6b, first note that the path along $x_1 = 0$ is not a strictly decreasing path. Any other path from $(0, 0.5)$ must include a segment of $(0, 0.5)$ to $(\epsilon_1, 0.5 + \epsilon_2)$ for some small $\epsilon_1, \epsilon_2$ with $\epsilon_1 \neq 0$. Then this segment is not strictly decreasing due to the alternating behavior of $\sin(-\frac{1}{|x_1|})$ when $x_1 = \epsilon_1 \neq 0$.

## 5.3 Proof of Proposition 2.1

($\Longrightarrow$) Let $x \in S$ denote a local minimum of $\phi \circ f$. There exists $\epsilon > 0$ such that $\phi(f(x)) \leqslant \phi(f(y))$ for all $y \in S \setminus \{x\}$ with $\|x - y\|_2 \leqslant \epsilon$. Since $\phi$ is increasing, it holds that $f(x) \leqslant f(y)$. Since $f$ is global, we deduce that $x$ is a global minimum of $f$, that is to say $f(x) \leqslant f(y)$ for all $y \in S \setminus \{x\}$. Since $\phi$ is increasing, it holds that $\phi(f(x)) \leqslant \phi(f(y))$ for all $y \in S \setminus \{x\}$. We conclude that $x$ is a global minimum of $\phi \circ f$.

($\Longleftarrow$) Simply apply the previous argument to $\phi^{-1} \circ (\phi \circ f)$, where $\phi^{-1}$ denotes the inverse of $\phi : f(S) \longrightarrow \phi \circ f(S)$.

## 5.4 Proof of Proposition 2.2

($\Longrightarrow$) Let $x' \in S'$ denote a local minimum of $f \circ \varphi^{-1}$. There exists $\epsilon' > 0$ such that $f(\varphi^{-1}(x')) \leqslant f(\varphi^{-1}(y'))$ for all $y' \in S' \setminus \{x'\}$ with $\|x' - y'\|_2 \leqslant \epsilon'$. Since $\varphi$ is continuous, there exists $\epsilon > 0$ such that $f(\varphi^{-1}(x')) \leqslant f(y)$ for all $y \in S \setminus \{\varphi^{-1}(x')\}$ with $\|\varphi^{-1}(x') - y\|_2 \leqslant \epsilon$. Hence, $\varphi^{-1}(x')$

(a) $f_2$

(b) $f_{1.5}$

(c) $f_{1.25}$

(d) $f_1$

Figure 5: Compact convergence of global functions implies that strict local minima are global

is a local minimum of $f$. Since $f$ is global, it holds that $f(\varphi^{-1}(x')) \leqslant f(y)$ for all $y \in S$. Since $\varphi$ is a bijection, $f(\varphi^{-1}(x')) \leqslant f(\varphi^{-1}(y'))$ for all $y' \in S'$, implying that $x'$ is a global minimum of $f \circ \varphi^{-1}$.

($\Longleftarrow$) Simply apply the previous argument to $(f \circ \varphi^{-1}) \circ \varphi$.

## 5.5 Proof of Proposition 2.5

One direction is obvious. For the other direction, we propose a proof by contrapositive. Let $X \subset S$ denote a local minimum that is not a global minimum. There exists $\epsilon > 0$ such that the uniform neighborhood $V := \{y \in S \mid \exists x \in X : \|x - y\|_2 \leqslant \epsilon\}$ satisfies $f(x) \leqslant f(y)$ for all $x \in X$ and for all $y \in V \setminus X$. Also, there exists $z \in S \setminus V$ such that $f(z) < f(x)$ for all $x \in V$. Since $f$ is continuous on the compact set $V$, it attains a minimum $x' \in V$ such that $f(z) < f(x')$. If $x' \in X$, then for all $y \in S$ such that $\|x' - y\|_2 \leqslant \epsilon$, it holds that $f(z) < f(x') \leqslant f(y)$. Thus, $x'$ is local minimum that is not a global minimum. If $x' \in V \setminus X$, then $f(x') \leqslant f(x) \leqslant f(x')$ for all $x \in X$. Consider a point $x \in X$. For all $y \in S$ such that $\|x - y\|_2 \leqslant \epsilon$, it holds that $f(x) = f(y)$ if $y \in X$ and $f(x) \leqslant f(y)$ if $y \notin X$. Together with the fact that $f(z) < f(x') = f(x)$, we deduce that $x$ is a local minimum that is not a global minimum.

(a) Global function devoid of a strictly decreasing path from $(0, 1/2)$ to a global minimizer

(b) Global function that is nowhere differentiable

Figure 6: Notable examples (with $x_1$-axis on the right and $x_2$-axis on the left)

## 5.6 Proof of Proposition 2.6

We propose a proof by contrapositive. Assume that $f$ is not constant on a local minimum $X \subset S$ that is not a global minimum. The minimum $X$ admits a uniform neighborhood $V$ such that $f(x) \leqslant f(y)$ for all $x \in X$ and for all $y \in V \setminus X$. Since $f$ is continuous on the compact set $V$, there exists $x' \in V$ such that $f(x') \leqslant f(x)$ for all $x \in V$. If $x' \in V \setminus \text{int}(X)$ where "int" stands for interior, then $f$ is constant on $X$ because $X$ is a local minimum. Therefore, $x' \in \text{int}(X)$ and $f(x') < f(x)$ for all $x \in \partial X := X \setminus \text{int}(X)$. Consider the compact set defined by $X' := \{x \in X \mid f(x') = f(x)\}$. The set $V$ satisfies $f(x) < f(y)$ for all $x \in X'$ and $y \in V \setminus X'$. Since $X' \subset X$, there exists a uniform neighborhood $V'$ of $X'$ satisfying $f(x) < f(y)$ for all $x \in X'$ and for all $y \in V' \setminus X'$. Hence, $X'$ is a strict local minimum that is not global. To conclude, $f$ is not a weakly global function.

## 5.7 Proof of Proposition 2.7

Consider a sequence of global functions $f_k$ that converge compactly towards $f$. Since $S \subset \mathbb{R}^n$ and $\mathbb{R}^n$ is a compactly generated space, it follows that $f$ is continuous. We proceed to prove that $f$ is a weakly global function by contradiction. Suppose $X \subset S$ is a strict local minimum that is not global minimum. There exists $\epsilon > 0$ such that the uniform neighborhood $V := \{y \in S \mid \exists x \in X : \|x - y\|_2 \leqslant \epsilon\}$ satisfies $f(x) < f(y)$ for all $x \in X$ and for all $y \in V \setminus X$. Since $f$ is continuous on the compact set $X$, it attains a minimal value on it, say $\inf_X f := \alpha + \inf_S f$ where $\alpha > 0$ since $X$ is not a global minimum. Consider a compact set $V \subset K \subset S$ such that $\inf_K f \leqslant \alpha/2 + \inf_S f$. Since $f$ is continuous on the compact set $\partial V$, it attains a minimal value on it, say $\inf_{\partial V} f := \beta + \inf_X f$ where $\beta > 0$ by strict optimality. Let $\gamma := \min\{\alpha/2, \beta\}$. For a sufficiently large value of $k$, compact convergence implies that $|f_k(y) - f(y)| \leqslant \gamma/3$ for all $y \in K$. Since the function $f_k$ is compact on $V$, it attains a minimum, say $z \in V$. Therefore,

$$f_k(z) \leqslant \gamma/3 + \inf_V f \leqslant \beta/3 + \inf_V f < 2\beta/3 + \inf_V f \tag{13}$$

$$\leqslant -\gamma/3 + \beta + \inf_V f \leqslant -\gamma/3 + \inf_{\partial V} f \leqslant \inf_{\partial V} f_k. \tag{14}$$

Thus, $z \in \text{int}(V)$. We now proceed to show by contradiction that $z$ is a local minimum of $f_k$. Assume that for all $\epsilon' > 0$, there exists $y' \in S \setminus \{x\}$ satisfying $\|x - y'\|_2 \leqslant \epsilon'$ such that $f_k(z) > f_k(y')$. We can choose $\epsilon'$ small enough to guarantee that $y'$ belongs to $V$ since $z \in \text{int}(V)$. The point $y'$ then contradicts the minimality of $z$ on $V$. This means that $z \in V$ is a local minimum of $f_k$. Now, observe that

$$\inf_K f_k \leqslant \gamma/3 + \inf_K f \leqslant \gamma/3 + \alpha/2 + \inf_S f \leqslant 2\alpha/3 + \inf_S f < 5\alpha/6 + \inf_S f \tag{15}$$

$$\leqslant \alpha - \gamma/3 + \inf_S f \;=\; -\gamma/3 + \inf_X f \;=\; -\gamma/3 + \inf_V f \;\leqslant\; \inf_V f_k \;\leqslant\; f_k(z). \qquad (16)$$

Thus, $z$ is not a global minimum of $f_k$. This contradicts the fact that $f_k$ is a global function.

## 5.8 Proof of Lemma 3.1

Based on the Clarke derivative [22, 23] for locally Lipschitz functions[5], the first-order optimality condition reads

$$0 \in \sum_{i_1,\ldots,i_{d-1}=1}^{n} x_{i_1} \ldots x_{i_{d-1}} \operatorname{sign}(x_{i_1} \ldots x_{i_{d-1}} x_i - y_{i_1} \ldots y_{i_{d-1}} y_i) \;\;,\quad i = 1,\ldots,n \qquad (17)$$

where

$$\operatorname{sign}(x) := \begin{cases} -1 & \text{if } x < 0, \\ [-1,1] & \text{if } x = 0, \\ 1 & \text{if } x > 0. \end{cases} \qquad (18)$$

If $y_i = 0$ for some $i \in \{1,\ldots,n\}$, then the above equations readily yield

$$0 \in \sum_{i_1,\ldots,i_{d-1}=1}^{n} x_{i_1} \ldots x_{i_{d-1}} \operatorname{sign}(x_{i_1} \ldots x_{i_{d-1}} x_i) = \operatorname{sign}(x_i) \sum_{i_1,\ldots,i_{d-1}=1}^{n} |x_{i_1} \ldots x_{i_{d-1}}| \qquad (19)$$

which implies $x_i = 0$. This reduces the dimension of the problem from $n$ to $n-1$, so without loss of generality we may assume that $y_i \neq 0$ for all $i = 1,\ldots,n$. After a division, observe that the following inclusion is satisfied:

$$0 \in \sum_{i_1,\ldots,i_{d-1}=1}^{n} |y_{i_1} \ldots y_{i_{d-1}}| \frac{x_{i_1} \ldots x_{i_{d-1}}}{y_{i_1} \ldots y_{i_{d-1}}} \operatorname{sign}\left( \frac{x_{i_1} \ldots x_{i_{d-1}}}{y_{i_1} \ldots y_{i_{d-1}}} t - 1 \right) \qquad (20)$$

for all $t \in \{x_1/y_1,\ldots,x_n/y_n\}$. Each term with $x_{i_1} \ldots x_{i_{d-1}} \neq 0$ in the above sum is an increasing step (set-valued) function of $t \in \mathbb{R}$ since it is the Clarke derivative of the convex function

$$g(t) = |x_{i_1} \ldots x_{i_{d-1}} t - y_{i_1} \ldots y_{i_{d-1}}|. \qquad (21)$$

The above sum is thus a increasing step function of $t \in \mathbb{R}$. Hence, the roots $x_1/y_1,\ldots,x_n/y_n$ all along belong to the same step. Jumps between the steps occur exactly at the following set of points:

$$\left\{ \frac{y_{i_1} \ldots y_{i_{d-1}}}{x_{i_1} \ldots x_{i_{d-1}}} \;\middle|\; i_1,\ldots,i_{d-1} \in \{1,\ldots,n\} \text{ and } x_{i_1} \ldots x_{i_{d-1}} \neq 0 \right\} \qquad (22)$$

This set is empty when $x = 0$; otherwise, none of its elements are equal to zero because $y \neq 0$. Given a jump point $\alpha \neq 0$ in the above set, the roots must therefore be all before or all after, that is to say:

$$\frac{x_1}{y_1},\ldots,\frac{x_n}{y_n} \leqslant \alpha \qquad \text{or} \qquad \alpha \leqslant \frac{x_1}{y_1},\ldots,\frac{x_n}{y_n}. \qquad (23)$$

We next prove that

$$\alpha > 0 \implies \frac{x_1}{y_1},\ldots,\frac{x_n}{y_n} \leqslant \alpha \qquad \text{and} \qquad \alpha < 0 \implies \alpha \leqslant \frac{x_1}{y_1},\ldots,\frac{x_n}{y_n}. \qquad (24)$$

Let us prove the first implication by contradiction. Assume that there exists $k \in \{1,\ldots,n\}$ such that $\alpha < x_k/y_k$. Since one root is after the jump point $\alpha$, all other roots are after the jump point $\alpha$. In particular, for all $i \in \{1,\ldots,n\}$, we have

$$0 < \alpha := \frac{y_{i_1} \ldots y_{i_{d-1}}}{x_{i_1} \ldots x_{i_{d-1}}} \leqslant \frac{x_i}{y_i} \qquad (25)$$

$$f^\circ(x;v) := \limsup_{\substack{y \to x \\ \lambda \downarrow 0}} \frac{f(y + \lambda v) - f(y)}{\lambda}$$

and the Clarke derivative is defined by $\partial f(x) := \{d \in \mathbb{R}^n : f^\circ(x;v) \geqslant \langle v, d \rangle, \quad \forall v \in \mathbb{R}^n\}$.

Therefore, all the roots are positive. By multiplying the above equation by the positive number $\frac{x_{i_1} y_i}{y_{i_1} x_i}$, we obtain

$$\frac{y_i y_{i_2} \cdots y_{i_{d-1}}}{x_i x_{i_2} \cdots x_{i_{d-1}}} \leqslant \frac{x_{i_1}}{y_{i_1}}. \tag{26}$$

Note that the left-hand side is a jump point, and the right-hand side is a root. Therefore, all the roots are after, and in particular:

$$\frac{y_i y_{i_2} \cdots y_{i_{d-1}}}{x_i x_{i_2} \cdots x_{i_{d-1}}} \leqslant \frac{x_i}{y_i}. \tag{27}$$

Again, since the roots are positive, by multiplying by $\frac{x_{i_2} y_i}{y_{i_2} x_i}$, we get

$$\frac{y_i^2 y_{i_3} \cdots y_{i_{d-1}}}{x_i^2 x_{i_3} \cdots x_{i_{d-1}}} \leqslant \frac{x_{i_2}}{y_{i_2}}. \tag{28}$$

Similarly, the left-hand side is a jump point, and the right-hand side is a root. Thus, all the roots are after, and in particular:

$$\frac{y_i^2 y_{i_3} \cdots y_{i_{d-1}}}{x_i^2 x_{i_3} \cdots x_{i_{d-1}}} \leqslant \frac{x_i}{y_i}. \tag{29}$$

Continuing this process, we ultimately obtain that

$$\frac{y_i^{d-1}}{x_i^{d-1}} \leqslant \frac{x_i}{y_i} \tag{30}$$

that is to say $1 \leqslant x_i/y_i$. If the inequality is an equality for all $i \in \{1, \ldots, n\}$, then $\alpha = 1 = x_k/y_k$, which is impossible since $\alpha < x_k/y_k$. Thus, there exists one root $t$ of (20) that is strictly greater than one. But this implies that every term in the sum in (20) is strictly positive, which is impossible. As a result, the first implication in (24) is true.

We next prove the second implication in (24) by contradiction. Assume that there exists $k \in \{1, \ldots, n\}$ such that $x_k/y_k < \alpha$. Since one root is before the jump point $\alpha$, all other roots are before the jump point $\alpha$. In particular, for all $i \in \{1, \ldots, n\}$, we have

$$\frac{x_i}{y_i} \leqslant \frac{y_{i_1} \cdots y_{i_{d-1}}}{x_{i_1} \cdots x_{i_{d-1}}} := \alpha < 0. \tag{31}$$

Therefore, all the roots are negative. Since $\alpha < 0$ is the product of $d-1$ negative terms, it must be that $d$ is even. Observe that $\frac{x_{i_1} y_i}{y_{i_1} x_i} > 0$ because it is a ratio of two roots. Now, similar to the case $\alpha > 0$, we obtain

$$\frac{x_{i_1}}{y_{i_1}} \leqslant \frac{y_i y_{i_2} \cdots y_{i_{d-1}}}{x_i x_{i_2} \cdots x_{i_{d-1}}}. \tag{32}$$

The right-hand side is a jump point, and the left-hand side is a root. Thus, all the roots are before, and in particular:

$$\frac{x_i}{y_i} \leqslant \frac{y_i y_{i_2} \cdots y_{i_{d-1}}}{x_i x_{i_2} \cdots x_{i_{d-1}}}. \tag{33}$$

Continuing this process (as in the case where $\alpha > 0$), we ultimately obtain that

$$\frac{x_i}{y_i} \leqslant \frac{y_i^{d-1}}{x_i^{d-1}}. \tag{34}$$

Since $d$ is even and $x_i/y_i < 0$, this implies that $x_i/y_i \leqslant -1$. If the inequality is an equality for all $i \in \{1, \ldots, n\}$, then $\alpha = -1 = x_k/y_k$, which is impossible since $x_k/y_k < \alpha$. Thus, there exists one root $t$ of (20) that is strictly less than $-1$. But this implies that every term in the sum in (20) is strictly negative, which is impossible. Consequently, the second implication in (24) holds.

Let us apply (24) to a root $x_{i_d}/y_{i_d}$ for some $i_d \in \{1, \ldots, n\}$:

$$\frac{y_{i_1} \cdots y_{i_{d-1}}}{x_{i_1} \cdots x_{i_{d-1}}} > 0 \implies \frac{x_{i_d}}{y_{i_d}} \leqslant \frac{y_{i_1} \cdots y_{i_{d-1}}}{x_{i_1} \cdots x_{i_{d-1}}} \quad \text{and} \quad \frac{y_{i_1} \cdots y_{i_{d-1}}}{x_{i_1} \cdots x_{i_{d-1}}} < 0 \implies \frac{y_{i_1} \cdots y_{i_{d-1}}}{x_{i_1} \cdots x_{i_{d-1}}} \leqslant \frac{x_{i_d}}{y_{i_d}}.$$

In both cases we find that

$$\frac{x_{i_1} \cdots x_{i_d}}{y_{i_1} \cdots y_{i_d}} \leqslant 1. \tag{35}$$

This inequality holds for all jump points (i.e. for all indices $i_1, \ldots, i_{d-1} \in \{1, \ldots, n\}$ such that $x_{i_1} \cdots x_{i_{d-1}} \neq 0$) and it is trivially true for all indices such that $x_{i_1} \cdots x_{i_{d-1}} = 0$. Therefore, (35) is true for all $i_1, \ldots, i_d \in \{1, \ldots, n\}$, which completes the proof of this lemma.

## 5.9  Proof of Lemma 3.2

When $x \in S$, notice that

$$
\begin{aligned}
f_1(x) &= \sum_{i_1,\ldots,i_d=1}^{n} |x_{i_1}\ldots x_{i_d} - y_{i_1}\ldots y_{i_d}| \\
&= \sum_{i_1,\ldots,i_d=1}^{n} |y_{i_1}\ldots y_{i_d}| \left| \frac{x_{i_1}\ldots x_{i_d}}{y_{i_1}\ldots y_{i_d}} - 1 \right| \\
&= \sum_{i_1,\ldots,i_d=1}^{n} |y_{i_1}\ldots y_{i_d}| - |y_{i_1}\ldots y_{i_d}| \frac{x_{i_1}\ldots x_{i_d}}{y_{i_1}\ldots y_{i_d}} \\
&= \left( \sum_{i=1}^{n} |y_i| \right)^d - \left( \sum_{i=1}^{n} |y_i| \frac{x_i}{y_i} \right)^d .
\end{aligned}
\tag{36}
$$

Given $\alpha > 0$, consider the function $\phi_\alpha : f_1(S) \longrightarrow \mathbb{R}$ defined by

$$
\phi_\alpha(t) = -\left[ -t - \left( \sum_{i=1}^{n} |y_i| \right)^d \right]^\alpha .
\tag{37}
$$

If $d$ is odd, then $\phi_\alpha$ is increasing when taking $\alpha = 1/d$. If $d$ is even, then $\phi_\alpha$ is increasing when $-t - (\sum_{i=1}^{n} |y_i|)^d$ is positive and $\alpha = 2/d$. Next, define the set

$$
S' := \{ x \in \mathbb{R}^n \mid x_{i_1}\ldots x_{i_d} \leqslant 1 , \quad \forall\, i_1,\ldots,i_d \in \{1,\ldots,n\} \}
\tag{38}
$$

and consider the homeomorphism $\varphi : S \longrightarrow S'$ defined by

$$
\varphi(x) = \left( \frac{x_1}{y_1}, \ldots, \frac{x_n}{y_n} \right) .
\tag{39}
$$

According to Proposition 2.1 and Proposition 2.2, $f$ is a global function on $S$, i.e. $f_1 \in \mathcal{G}(S)$, if and only if $\phi_\alpha \circ f_1 \circ \varphi^{-1}$ is a global function on $S'$. Thus, when $d$ is odd, $f_1 \in \mathcal{G}(S)$ if and only if $f_{\text{odd}}(x) := \phi_{1/d} \circ f_1 \circ \varphi^{-1}(x) = -\sum_{i=1}^{n} |y_i| x_i \in \mathcal{G}(S')$. When $d$ is even, $f_1 \in \mathcal{G}(S)$ if and only if $f_{\text{even}}(x) := \phi_{2/d} \circ f_1 \circ \varphi^{-1}(x) = -\left( \sum_{i=1}^{n} |y_i| x_i \right)^2 \in \mathcal{G}(S')$.

Consider the case when $d$ is odd. For all $i_1,\ldots,i_d \in \{1,\ldots,n\}$, define the constraint function $g_{i_1,\ldots i_d}(x) := x_{i_1}\ldots x_{i_d} - 1$. If $x_1\ldots x_n \neq 0$, then for any $i_1,\ldots,i_d \in \{1,\ldots,n\}$, it satisfies

$$
\nabla\, g_{i_1,\ldots i_d}(x) = \begin{pmatrix} N(1, i_1,\ldots i_d)/x_1 \\ \vdots \\ N(n, i_1,\ldots i_d)/x_n \end{pmatrix} x_{i_1}\ldots x_{i_d}
\tag{40}
$$

where $\nabla\, g_{i_1,\ldots i_d}(x)$ denotes the gradient of $g_{i_1,\ldots i_d}$ at $x$ and $N(i, i_1,\ldots, i_d)$ denotes the number of indices among $i_1,\ldots,i_d$ that are equal to $i$. If the constraint $g_{i_1,\ldots i_d}(x) \leqslant 0$ is active, then

$$
-x^T \nabla g_{i_1,\ldots i_d}(x) = -\sum_{k=1}^{n} N(k, i_1,\ldots i_d) < 0.
\tag{41}
$$

The Mangasarian-Fromovitz constraint qualification thus holds. A local minimum $x \in \mathbb{R}^n$ for the problem $\inf_{x \in S'} f_{\text{odd}}(x)$ must therefore satisfy the Karush-Kuhn-Tucker conditions:

$$
\begin{cases}
\displaystyle\sum_{\substack{i_1,\ldots,i_d=1 \\ N(i, i_1,\ldots, i_d) \neq 0}} \lambda_{i_1,\ldots,i_d} N(i, i_1,\ldots, i_d) \frac{x_{i_1}\ldots x_{i_d}}{x_i} = |y_i| , \quad \forall\, i \in \{1,\ldots,n\} , \\[2em]
x_{i_1}\ldots x_{i_d} \leqslant 1 , \quad \forall\, i_1,\ldots,i_d \in \{1,\ldots,n\} , \\[1em]
\lambda_{i_1,\ldots,i_d} \geqslant 0 , \\[1em]
\lambda_{i_1,\ldots,i_d}(x_{i_1}\ldots x_{i_d} - 1) = 0 .
\end{cases}
\tag{42}
$$

Here $\lambda_{i_1,\ldots,i_d} \geq 0, i_1, \ldots, i_d \in \{1, \ldots, n\}$, are the Lagrange multipliers. If $x_i \neq 0$ for some $i \in \{1, \ldots, n\}$, then, by complementarity slackness, the first line yields

$$0 \quad < \quad |y_i| \quad = \frac{1}{x_i} \sum_{\substack{\lambda_{i_1,\ldots,i_d} > 0 \\ N(i, i_1, \ldots, i_d) \neq 0}} \underbrace{\lambda_{i_1,\ldots,i_d} N(i, i_1, \ldots, i_d)}_{>0} \tag{43}$$

which implies that $x_i > 0$. As a result, $x \geqslant 0$. Together with feasibility, it results that $0 \leqslant x_i^d \leqslant 1$, leading to the inequalities $0 \leqslant x_i \leqslant 1$ for all $i \in \{1, \ldots, n\}$. Following Proposition 2.3, $f_{\mathrm{odd}}$ is thus global on $S'$ if $f_{\mathrm{odd}} \in \mathcal{G}(S'')$ where

$$S'' := \{\, x \in \mathbb{R}^n \mid 0 \leqslant x_i \leqslant 1 \,, \quad \forall\, i \in \{1, \ldots, n\} \,\} \tag{44}$$

which contains the global minimizer 0. From the notion of global functions, $f_{\mathrm{odd}} \in \mathcal{G}(S'')$ if the problem

$$\inf_{x \in \mathbb{R}^n} \quad -\sum_{i=1}^{n} |y_i| x_i \quad \text{subject to} \quad 0 \leqslant x_i \leqslant 1 \,, \quad \forall\, i \in \{1, \ldots, n\} \tag{45}$$

has no spurious local minima, which is obvious because the problem is a linear program.

Consider the case when $d$ is even. Since a feasible point $x \in S'$ satisfies $x_i^d \leqslant 1$, it must be that $-1 \leqslant x_i \leqslant 1$. Conversely, any point such that $-1 \leqslant x_i \leqslant 1$ belongs to $S'$. This implies that

$$S' := \{\, x \in \mathbb{R}^n \mid -1 \leqslant x_i \leqslant 1 \,, \quad \forall\, i \in \{1, \ldots, n\} \,\} \,. \tag{46}$$

According to Proposition 2.4, $f_{\mathrm{even}}(x) \in \mathcal{G}(S')$ if $f_{\mathrm{even}}(x)$ is a global function on both sets $S' \cap \{x \in \mathbb{R}^n | \sum_{i=1}^{n} |y_i| x_i \geq 0\}$ and $S' \cap \{x \in \mathbb{R}^n | \sum_{i=1}^{n} |y_i| x_i \leq 0\}$, and $f_{\mathrm{even}}(x)$ takes the same optimal value on both sets (the latter is obvious using symmetry). Using Proposition 2.1 again, we find that $f_{\mathrm{even}}(x)$ is a global function on these two sets if and only if

$$-\sum_{i=1}^{n} |y_i| x_i \in \mathcal{G}\left( S' \cap \left\{ x \in \mathbb{R}^n | \sum_{i=1}^{n} |y_i| x_i \geq 0 \right\} \right) \tag{47}$$

and

$$\sum_{i=1}^{n} |y_i| x_i \in \mathcal{G}\left( S' \cap \left\{ x \in \mathbb{R}^n | \sum_{i=1}^{n} |y_i| x_i \leq 0 \right\} \right) \,, \tag{48}$$

which are true because they are associated with the following linear programs:

$$\inf_{x \in \mathbb{R}^n} \quad -\sum_{i=1}^{n} |y_i| x_i \quad \text{subject to} \quad \begin{cases} -1 \leqslant x_i \leqslant 1 \,, & \forall\, i \in \{1, \ldots, n\} \\ \sum_{i=1}^{n} |y_i| x_i \geqslant 0. \end{cases} \tag{49}$$

and

$$\inf_{x \in \mathbb{R}^n} \quad \sum_{i=1}^{n} |y_i| x_i \quad \text{subject to} \quad \begin{cases} -1 \leqslant x_i \leqslant 1 \,, & \forall\, i \in \{1, \ldots, n\} \\ \sum_{i=1}^{n} |y_i| x_i \leqslant 0. \end{cases} \tag{50}$$