[Reviews · NeurIPS 2018]

Reviewer 1



Summary: This paper studies the problem of absence of spurious local optimality for a class of non-convex optimization problems. Building on nice properties of the so-called global functions, the authors show that a class of non-convex and non-smooth optimization problems arising in tensor decomposition applications have no spurious local minimum. Comments: The paper discusses several properties of global functions, which are nice. Here I have some comments on the paper: 1. Although in the introduction the authors mention matrix completion and sensing as the motivation of the paper, it seems that the main result of this paper, i.e., Proposition 1.1, does not really deal with the missing data related problems. Indeed, the optimization problems in Proposition 1.1 are an extension of low-rank approximation to the tensor format with non-smooth objective, but requiring that the decomposition has the right rank as the given matrix and the given matrix is fully observed. This indeed is less interesting comparing with missing data cases, having fewer applications in reality. 2. The paper claims that the l1 norm can handle outlier. But dosen't problem (8) has the same global solution as the quadratic loss? --- 0 will be the optimal loss for both problems. So what is the advantage of studying l1 norm loss compared with studying quadratic loss, as all previous works have done? ================ I read the authors' rebuttal and I think the authors agree my concern in the first round of review. Please address them in the further revision of this paper.

Reviewer 2



This paper studies the condition for absence of spurious optimality. In particular, the authors introduce 'global functions' to define the set of continuous functions that admit no spurious local optima (in the sense of sets), and develop some corresponding definitions and propositions for an extending characterization of continuous functions that admit no spurious strict local optima. The authors also apply their theory to l1-norm minimization in tensor decomposition. Pros: In my opinion, the main contribution of this paper is to establish a general math result and apply it to study the absence of spurious optimality for a specific problem. I also find some mathematical discoveries on global functions interesting, which include: -- In section 2, the paper provides two examples to show that: (i). there exist global functions that are nowhere differentiable; (ii). there may not exist a strictly decreasing path from any point to a global minimum for a coercive global function defined on compact sets. These two non-trivial examples imply that 'global function' may have bad properties. -- From definition 2.4 to definition 2.6, the paper introduces 'minimum in the sense of sets' and 'weakly global functions'. These definitions are non-trivial in the sense that the crucial property of 'all strict local minima are global minima' is not consistent in the sense of points and sets. This non-equivalence means that the set of 'weakly global functions' is a proper subset of functions whose strict local minima are all global minima, which implies that we may expect more detailed characterization of weakly global functions. Cons: There are a few places that can be improved. See detailed comments below. (1) Although the mathematical tools and examples are interesting, the studied problem is not as interesting as the math part. Often we expect a powerful theory to solve a general class of problems, but I find it a bit disappointing that this paper only solves a relatively narrow and less interesting problem. The significance of the main general theorem Proposition 2.7 would be much more appreciated if there are other interesting examples. (2) Algorithm. In Line 61, the paper claims that one can minimize the function in proposition 1.1 using a local search algorithm. Which local search algorithm can guarantee local optimality for a non-smooth function? To my knowledge, at least subgradient method and coordinate descent method can not. It is clearly an issue that classical methods cannot guarantee local optimality in general. Thus proving a non-smooth function is a global function is not enough. Whether this is a big issue or not is debatable, but it is misleading to claim local minima of non-smooth functions are easy to find in general. (3) Some comments on the wording. (i) The title “A theory on the absence of spurious optimality” is a bit too general. This is not the first theoretical paper that studies spurious optimality, and not the first one that attempts a general theory. It is also not a comprehensive study. Something more specific may be more accurate (perhaps something related to Prop. 2.7?) (ii) The authors claim “This is the first result concerning nonconvex methods for nonsmooth objective functions”. This claim is questionable as there are quite a few previous papers that analyze neural networks with ReLU activations. (iii) The statement “It has received a lot of attention recently due to the discovery in [25, 10] stating that the problem generally admits no spurious local minima (i.e. local minima that are not global minima)” is not true. [25] requires extra regularizers (which is crucial), and [10] requires RIP assumption. But the statement seems to imply (2) is a global function, given that the whole paper is about global functions. This is misleading. In fact, without extra restrictions (2) can be a non-global function. (4) Some technical comments. --The description of proposition 2.4 is a bit confusing. According to definition 2.4, the global minimum for a set is defined with respect to a particular function f. However, the conclusion focuses on the set of global functions over some particular sets. The assumption and conclusion don't match perfectly. I guess what you intend to claim is that: if f is a global function on X_alpha for some alpha in some index set A, and X_alpha are global minima of f on the union of X_alpha, then f is a global function on the union of X_alpha. This claim is true. --A point in the proof of lemma 3.2 and proposition 2.3: to utilize proposition 2.3, you need to show that S'' is a global minimum of f_odd on S'. This is not proven in the proof. However, I guess this is implicitly true due to the continuity of f_odd. Correspondingly, is assumption 1 really needed in proposition 2.3? (5) Some typos. -- Line 411: alpha should be alpha/2. -- Line 473: the term in the beginning should be corrected. Same is it with equation (37). -- In the first equation of (42): 0 in the right hand side should be |yi|. --Equation (4): It is not the clear to me where matrices A in (2) are. Do you assume here that A=I?

Reviewer 3



Summary: The paper studies a class of functions which admit no spurious minima. First, the authors explore various closure-related properties such as post-composition of monotonically increasing functions, uniform convergence and 'decreasing-paths', to name a few. Next, mainly motivated by the absence of closure property under uniform convergence, the authors generalize the notion of extremality for sets which allows taking uniform limits (over compact sets). Lastly, this result is applied for tensor-related optimization problems and is used to prove the absence of generalized (strict) local minima. Evaluation: Existence results of optimization manifolds without spurious minima is now a very active research area. The paper mainly contributes to this area by showing that global functions (as defined in the paper) do not exhibit the 'decreasing-paths' property, and finding a relaxed notion of minima under which one can show closure property of taking uniform limits of global functions. This provides a nice way of expanding a known set of global functions in a relatively systematic way (but seems to be less effective in establishing 'globality' of a given function). The paper is well-written and very nice and easy to follow. Comments: Consider providing a more explicit definitions for: 'fully observed' (L41), 'uniform neighborhood' and 'Clarke derivative' L93 - 'cosinus' -> cosine? L98 - figure 4a shoud be 4b Paragraph in the end of page 3 - 'strictly strictly' L104 - To me, the fact that f(x_1,x_2) admit no strictly decreasing path from (0,0.5) wasn't entirely clear from figure 6a. Can you provide a somewhat more analytic proof for this?